# Widening East-West inequality in life expectancy in Europe during the COVID-19 pandemic: An international comparative study

Vladimir M. Shkolnikov[1,2], Sergey Timonin[3,4*], Dmitri Jdanov[1], Naomi Medina-Jaudes[5], Nazrul Islam[6], David A. Leon[5]

1 Laboratory of Demographic Data, Max Planck Institute for Demographic Research, Rostock, Germany, 2 International Laboratory for Population and Health, National Research University Higher School of Economics, Moscow, Russia, 3 School of Demography, The Australian National University, Canberra, Australia, 4 National Centre for Epidemiology and Population Health, The Australian National University, Canberra, Australia, 5 Department of Noncommunicable Disease Epidemiology, London School of Hygiene & Tropical Medicine, London, United Kingdom, 6 Primary Care Research Centre, Faculty of Medicine, University of Southampton, Southampton, United Kingdom

* sergey.timonin@anu.edu.au

## Abstract

For decades, life expectancy in Eastern Europe has lagged behind that in Western Europe, although the gap has narrowed since the mid-2000s. We examined the impact of the COVID-19 pandemic on these long-term trends by estimating and decomposing East-West differences in excess mortality and life expectancy losses. Regression models were used to explore the contribution of multiple factors to the observed differences. While excess mortality was initially low in Eastern Europe, partly due to lower air connectivity, by October 2020 and throughout 2021, the region experienced significantly higher life expectancy losses than Western Europe. These differences could not be attributed to greater frailty in the Eastern European populations. Rather, lower vaccination coverage and less trust in government in Eastern Europe explained about half of the gap in 2021. After the peak of the pandemic, the East-West gap in life expectancy began to narrow again. Our findings highlight that the factors behind the East-West differences in the timing and severity of pandemic-related mortality may have their ultimate origins in differences between societies that were established during the Cold War. Strengthening public health infrastructure and social trust should be considered to mitigate the unequal impact of future crises.

## Introduction

From the late 1960s, life expectancy in the former communist countries of Eastern Europe lagged behind those of Western Europe, but from 2000 this gap started to narrow [1]. The COVID-19 pandemic in 2020–21 was equally unexpected for all countries. Higher or lower losses of human lives in these years reflected the ability

**Data availability statement:** The raw data originated from the open access sources listed in the References. We also provide figures and tables data in separate Excel files (https://github.com/VMSdemo/East-West-contrast-in-life-expectancy-losses-in-2020-21). Some of the Excel files contain calculations (including decomposition of life expectancy losses) and final data manipulations.

**Funding:** ST acknowledges support from the Australian Research Council (DP210100401). NI acknowledges support from the UK National Institute for Health and Care Research (HDRUK2022.0313). There was no additional external funding received for this study. The funders had no role in study design, data collection and analysis, decision to publish, or preparation of the manuscript.

**Competing interests:** The authors have declared that no competing interests exist.

of different governments, societies, and populations to respond to the emergence of a major health challenge. As previously reported, the overall levels of pandemic excess mortality in Eastern European countries in 2020–21 were higher than in Western countries [2–10]. Schöley and colleagues noted that in Europe, the East-West differences in life expectancy losses were larger in 2021 than in 2020 [10]. They also found that life expectancy losses in the fourth quarter of 2021 were correlated with vaccination coverage. Inter-country differences in 2021 are of special interest because in principle vaccines became available everywhere by February-March 2021.

In addition to establishing differences in overall excess mortality between East and West, some studies have noted that countries of the East had a relatively mild first phase of the pandemic, with excess deaths only climbing steeply from autumn 2020 onwards [11]. In contrast, many Western countries were hit hard in the spring of 2020 but managed to avoid a large increase in the autumn/winter of 2020 and early 2021 [9–10]. The end of the acute phase of the pandemic crisis situation of 2021 came in 2022 after the emergence of the Omicron variant, with its lower case fatality [12]. This, together with increased levels of immunity, resulted in a weakening of the mortality impact of COVID-19 and the increasing importance of other causes of death [13–14].

The overarching aim of the work reported in this paper is to explore why the differential impact of the pandemic in Europe echoed the historical life expectancy divide between the former communist countries of the East and the rest of Europe. Apart from differences in vaccination coverage between the East and West, which would only have impacted the trajectory of the pandemic from 2021 onwards, several other factors have been mentioned as explaining East-West differences in pandemic mortality. Inadequate or ineffective non-pharmaceutical interventions (NPIs), as well as low trust in government and science have been considered as central [15]. Individual-level evidence from Germany linked low trust among older individuals with exposure to communism [16]. The lower ability of Eastern European authorities to enforce regulatory measures may also have contributed to higher life expectancy losses in the East [17]. Differences in pre-pandemic levels of air travel connectivity between European countries could also be important, although this has not been previously explored as an explanation of the timing of the pandemic's impact in Eastern compared to Western Europe.

## Materials and methods

### Data source

We used annual mortality data for European countries (n = 28) up to and including 2023 from the Human Mortality Database (HMD) [18] and weekly mortality data from the Short-Term Mortality Fluctuations (STMF) series [19] (S1-S6 Appendix). We divided the 28 populations into two country groups: Eastern Europe (referred to as East), consisting of 11 former communist countries and Western Europe (referred to as West), consisting of the remaining 17 countries (S1 Table).

## Statistical analysis

To provide a broader context, we first looked at life expectancy trends across Europe in 2000−2023. However, the primary focus of our main analyses was on excess mortality and life expectancy losses in 2020 and 2021, the years of maximal impact of COVID-19 on mortality in Europe.

Our analyses of differences between Eastern and Western European countries in 2020−21 included: 1) assessment of excess mortality by week to determine the dynamics of the pandemic by country and region; 2) correlation between excess mortality in spring 2020 and pre-pandemic levels of flight connectivity; 3) quantification and age decomposition of life expectancy losses in 2020 and 2021; 4) analysis of associations between life expectancy losses in 2021 and selected factors across the countries.

**Differences in weekly excess mortality.** Temporal differences in excess mortality between countries were characterized using the month of the first major excess death rate (EDR) peak and the total EDR for 2020−21, and maps were used to visualize these patterns. See Supplementary Data and Methods (S1-S6 Appendix) for detailed explanation of the methods for analysis of weekly mortality data.

**Air connectivity and excess mortality.** We examined the likelihood of the COVID-19 case importation into each country in the immediate pre-pandemic period using an index of air passenger connectivity. We hypothesized that the number of such imported cases and subsequent excess mortality might be related to each country's travel connectivity to other countries in the region. We assessed the rank order correlations for European countries between EDRs in March-April (weeks 10–18) 2020 and the average daily number of arriving flights from other European countries in the period February 10–23, 2020, using data from the OpenSky network [20]. See S1-S6 Appendix for more details.

**Life expectancy losses and age decomposition.** We compared the observed life expectancies in 2020 and 2021 with the expected values predicted from the past mortality trends, i.e., the life expectancies that would have been observed in the absence of the pandemic (S1 Fig). We used the Lee-Carter model to make the projection using the 2005–19 baseline period, based on previous sensitivity analyses of the choice of baseline period [8–19- 21]. Life expectancy losses were defined as the differences between the observed and expected life expectancies in 2020 and 2021. We then decomposed the losses into the contribution of two broad age groups: 0–64 years and 65+years. See S4 and S5 Appendix for details and equations.

We also quantified the contributions of the baseline mortality (pre-pandemic death rates) and the relative (proportional) mortality change (the pandemic *per se*) to the East-West difference in life expectancy losses. A counterfactual approach was used to calculate the corresponding Level and Change components [22]. See S5 Appendix for details and equations.

**Associations between life expectancy losses and factors.** We examined the potential contribution of population-level factors to explaining the very large East-West gap in life expectancy losses in 2021. These factors were the stringency index, vaccination coverage, trust in government, trust in science, and a regulatory enforcement score [23–26]. (See S6 Appendix for a detailed description of the explanatory variables).

Our analysis was guided by a simple conceptual framework (Fig 1) relating different factors to life expectancy losses in 2021. Proxy data for a number of these factors (marked in green in Fig 1) were used in a meta-regression framework. We checked the normality of the distributions and performed linearity and heteroscedasticity tests for the explanatory variables. We then calculated Pearson's *r* for the correlations of each of these variables with male and female life expectancy losses in 2021.

Finally, we regressed sex-specific life expectancy losses on the East-West dummy to assess the overall East-West difference. To see how this difference was attenuated in response to adjustment for explanatory variables, we ran additive regression models including the East-West dummy and one or two explanatory variables.

Additional methodological details are provided in the S1-S6 Appendix.

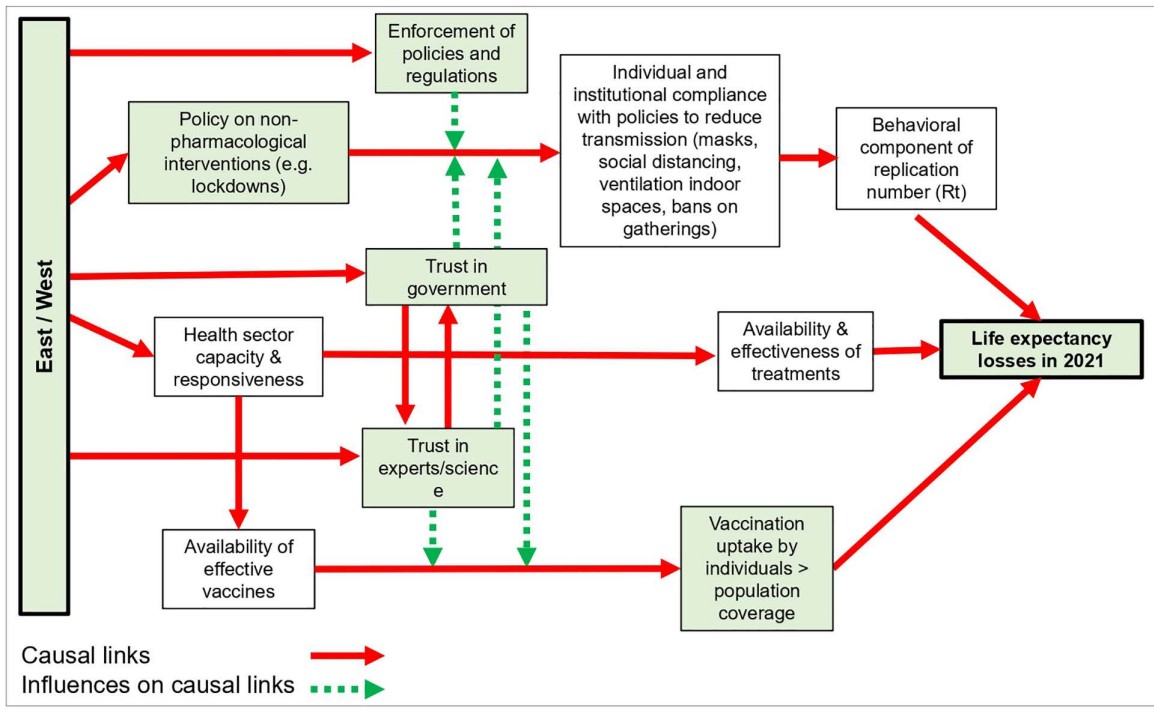

**Fig 1. Conceptual diagram: factors that may explain the East-West gap in life expectancy losses in 2021.** Note: The green color of the boxes indicates the availability of numerical proxy variables.

*Software.* Life table calculations were carried out in R (version 4.2.2); the rest of the statistical analysis was conducted in Stata 17. The scripts and corresponding data and calculations are available at: https://github.com/VMSdemo/East-West-contrast-in-life-expectancy-losses-in-2020-21. Maps were built in ArcGIS 10.8.2.

## Results

### Trends in life expectancy in 2000−23

The long-term life expectancy disadvantage of the East compared to the West began to reduce in the late 1990s in the new EU member states of Central and Eastern Europe, and in the mid-2000s in Russia (S2A and S2B Figs). Between 2000 and 2019, the difference in average life expectancies between the East and West decreased from 7.7 years to 6.3 years for males, and from 4.4 years to 3.2 years for females. This convergence was reversed by the COVID-19 pandemic: in 2020, the East-West difference increased to 6.5 years for males and 3.5 years for females. In 2021, the gap widened further to 7.9 and 4.9 years, respectively. This was caused by a slightly larger fall in life expectancy in the East than in the West in 2020 (males: 0.9 years in Eastern Europe vs 0.7 in Western Europe; females: 0.8 years and 0.5 years, respectively). In 2021, the East saw a continuing fall of 1.3 years for both males and females, while the West experienced a small rebound with a gain of 0.1 years in both sexes. All Eastern European countries had larger declines in 2021 than in 2020.

In 2022−23, all countries experienced a post-pandemic recovery that was steeper in the East than in the West. Male and female life expectancies increased by 2.6 and 2.5 in the East and by 0.7 and 0.6 years in the West. The East-West life expectancy gaps in 2023 were already lower than those in 2019 by 0.5 years. Nevertheless, life expectancies in 2023 in most Eastern and Western countries were still below the levels expected from pre-pandemic trends and the East-West gap was still larger than the expected one.

## Patterns of excess mortality in 2020−21

The weekly EDRs in 2020−21 by country are shown in Fig 2, together with the Eastern and Western mean EDRs. The countries with the highest EDR peaks in the first pandemic wave (March-May 2020) were in Western Europe. During this initial phase, the pandemic had little impact on mortality in countries of the East, except Russia, which experienced moderate excess mortality in May-June 2020. This changed in October 2020 when most Eastern countries experienced marked increases in weekly excess deaths. From November 2020 to January 2021, the EDR peak was higher and wider in the East than in the West. Most importantly, countries of the East experienced further massive elevations of weekly EDRs throughout 2021.

Panel A of Fig 3 shows that the first major peak in EDR in each country moved over time from the southwest to the northeast of Europe. The earliest major excess mortality waves in the spring of 2020 occurred in Italy, Spain, England and Wales, Scotland, Northern Ireland, Belgium, the Netherlands, France, and Sweden. Most other Western countries and nearly all Eastern ones experienced later major peaks between November 2020 and January 2021.

Despite the earlier onset of excess mortality and its higher level in spring 2020 in many Western countries, total mortality excess in 2020−21 was substantially higher in Eastern countries than in Western ones, except for Slovenia (Fig 3, Panel B). It is particularly striking that while the Western countries situated in the center of Europe, such as Germany, Austria, and Switzerland, had similar timings in their first pandemic peaks as neighboring countries in the East (Poland, Czechia, Hungary, and others), their overall rate of excess mortality in 2020−21 was much lower. Even Latvia and Estonia, which had very late excess mortality onset in October-November 2021, similarly to Norway and Finland, experienced much higher total excess mortality in 2020−21 than any Western country.

## Association between flight connectivity and excess mortality in spring 2020

S2 Table demonstrates the higher intensity of arriving flights in Western countries compared to Eastern European countries. In the immediate pre-pandemic period (10–23 February 2020), the mean number of flights per day arriving in all Western European countries from another European country was 6356, compared to only 816 in all Eastern countries.

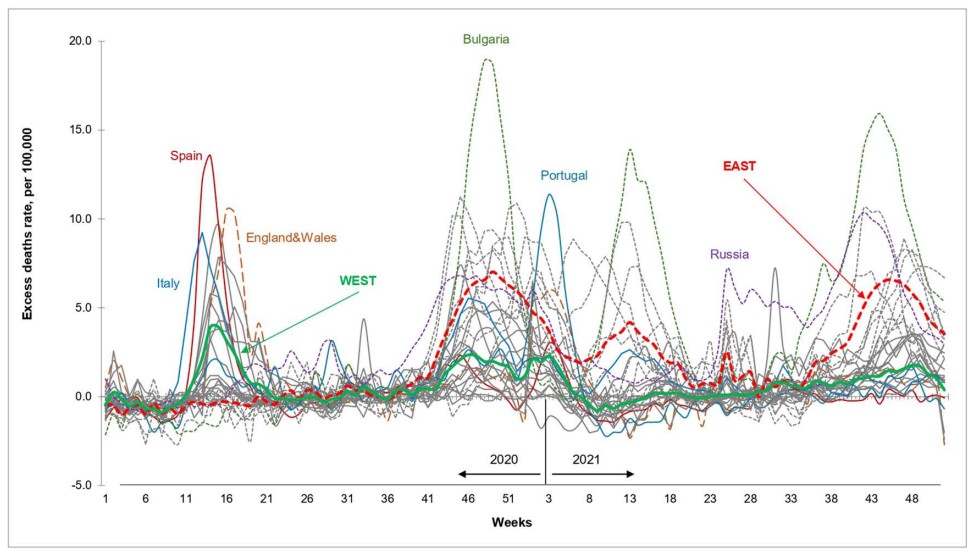

**Fig 2. Weekly excess death rates in 2020 and 2021, total population.** Note: Data shown in this Figure is provided at https://github.com/VMSdemo/East-West-contrast-in-life-expectancy-losses-in-2020-21.

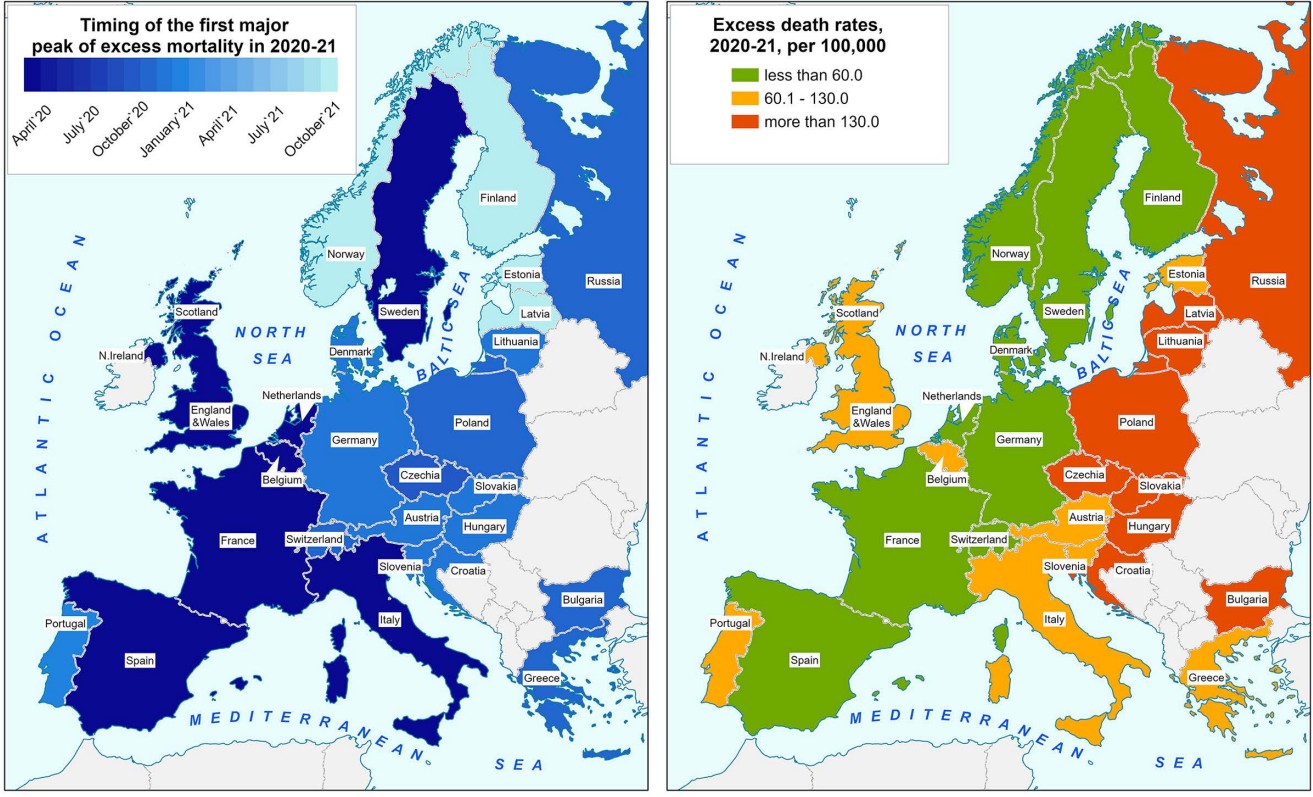

**Fig 3. Spatial pattern of excess mortality during the COVID-19 pandemic: Timing of the first major peak (left panel) and Excess death rates in 2020−21 (right panel).** Contains information from OpenStreetMap and OpenStreetMap Foundation, which is made available under the Open Database License. Note: Data shown in this Figure is provided at https://github.com/VMSdemo/East-West-contrast-in-life-expectancy-losses-in-2020-21.

S3 Fig suggests a positive relationship between the excess death rate in March-April 2020 and the flight connectivity in mid-February 2020, with Spearman *rho* of 0.782 (p < 0.001) (95% CI 0.627–0.936). The main outlier is Germany, which is exceptional as Frankfurt is by far its busiest airport and one of the largest transit hubs in the world, but EDR in Germany was far lower than that in the UK, even though they have similar connectivity.

**Life expectancy losses in 2020−21 and the re-emergence of the East-West divide.** Life expectancy losses in 2020 overlapped between East and West (S3 Table). However, in 2021, the losses were substantially higher in the East (2.9 years (95%CI 2.7–3.1) for males and 2.5 years (2.3–2.7) for females) than in the West (1.0 years (0.8–1.2) for males and 0.7 years (0.5–0.9) for females) (see also S4 Fig). Notably, the mean life expectancy losses in Western Europe in 2021 were below the minimum country-specific losses in Eastern countries in both sexes (S3 Table).

As in earlier studies [8–10], the estimated life expectancy losses in 2020−21 were higher for males than for females. The male-female difference in life expectancy losses was about the same in the East compared to the West, with its mean values of a quarter of a year in 2020 and close to 0.4 years in 2021. In 2021, Russia and Bulgaria constituted an interesting exception from this regularity with the male-female gap of about 0.5 years in Russia and 0.1 years in Bulgaria.

While there was already a tendency for higher life expectancy losses in Eastern countries in 2020, this became more pronounced in 2021. The upper panels of Fig 4 show that countries with the largest life expectancy losses in 2020 included both Western (Italy, Spain, parts of the UK, and Belgium) and Eastern (Russia, Lithuania, Bulgaria, Poland, Czechia) countries. However, the lower panels of Fig 4 demonstrate that in 2021, the ten countries with the highest life expectancy losses were all from Eastern Europe.

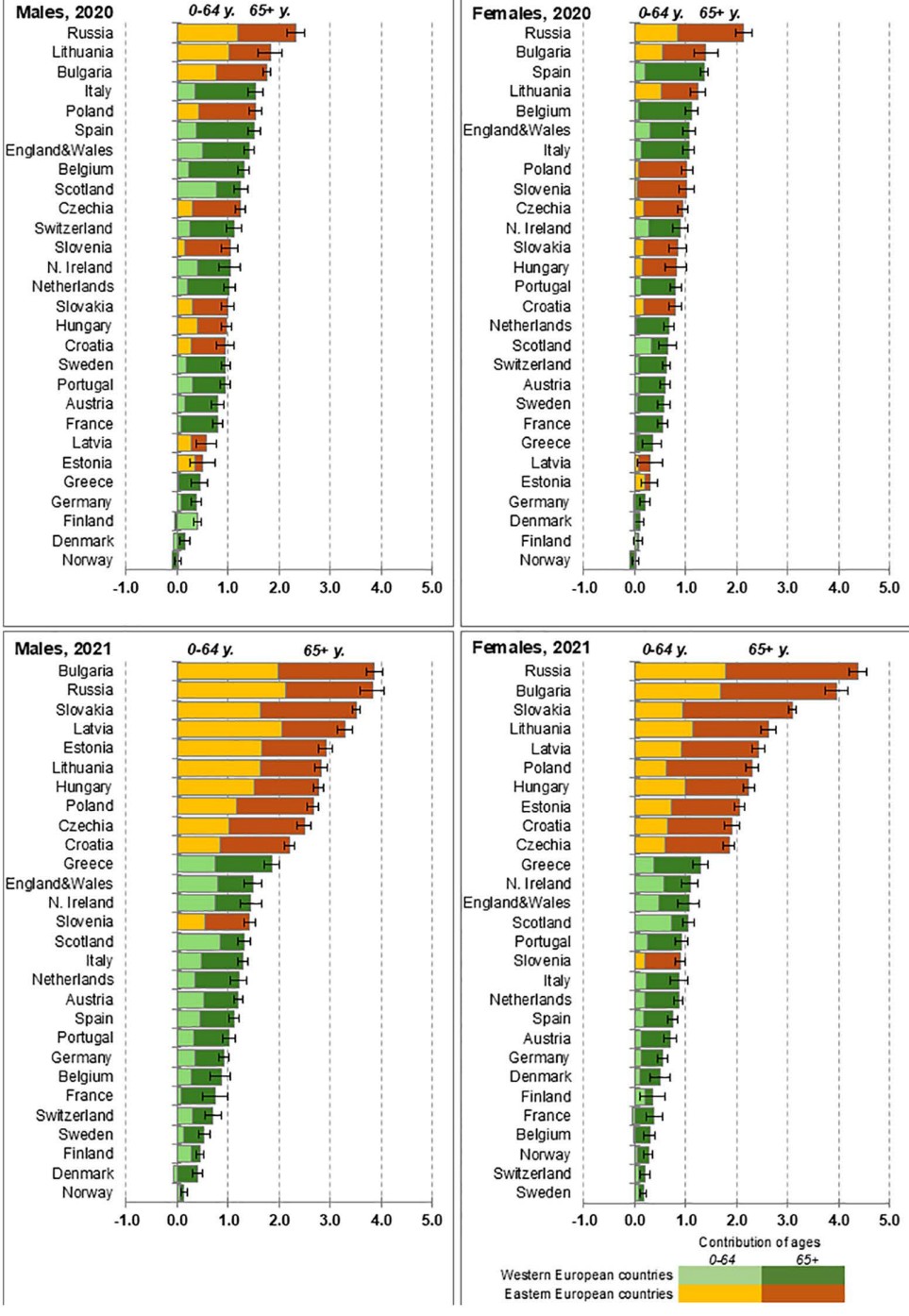

**Fig 4. Life expectancy losses and their age components, by year and sex, in years.** Note: Data shown in this Figure is provided at https://github.com/VMSdemo/East-West-contrast-in-life-expectancy-losses-in-2020-21.

Fig 4 also shows the contribution of different age groups to overall losses for each country. Overall, in 2020, the contributions of excess deaths occurring under the age of 65 years to total life expectancy losses were 31.5% for males (25.3% in the West vs. 39.5% in the East) and 23.2% for females (19.3% in the West vs. 28.5% in the East). In 2021, deaths under 65 years made up 44.1% of the total losses of life expectancy for males (40.3% in the West vs. 49.7% in the East) and 35% for females (34.8% in the West vs. 35.2% in the East).

**Do baseline differences in mortality explain the East-West gap in 2021?** The East-West differences in the impact of the COVID-19 pandemic on excess mortality and life expectancy losses can be separated into those due to the greater proportional increase in mortality in the East compared with the West during the pandemic, and those due to the higher and more premature baseline mortality in the East, reflecting the poorer pre-pandemic health status of the Eastern European populations.

The average difference in the life expectancy losses in 2021 between the East and the West was about 2 years for males and females (S3 Table). As shown in S4 Table, the contribution of the larger relative excess mortality in the East during the pandemic to these differences was 88% for men and 94% for women. Further evidence for a substantial East-West difference in relative excess mortality is provided in S5 Fig.

**Associations of factors with East-West differences in life expectancy losses in 2021.** The scatterplots in S6 Fig show statistically significant correlations between life expectancy losses in 2021 and all individual country-level factors (explanatory variables), except for the stringency index. Table 1 shows the effect of adjusting for several factors on the East-West difference in life expectancy losses in 2021. Without adjustment for any factors, the life expectancy losses were 2 years (males) and 1.9 years (females) higher in the East compared to the West. Vaccination coverage as of September 1, 2021, led to the largest attenuation of this effect of any single variable (down to 1.5 years for males and 1.2 years for females). Regulatory enforcement and trust in government had the second and third largest effects on the East-West gap, respectively. The stringency index of NPIs did not attenuate the association at all. Vaccination combined with trust in government had the largest reduction of any model, halving the East-West gap in life expectancy losses. Adjustment for vaccination together with regulatory enforcement had a similar effect for women, but a slightly smaller attenuating effect for men. The two-factor models had a marginally better fit than the single-factor (vaccination) models.

**Table 1. The model-based East-West difference in life expectancy losses in 2021, in years.**

| Covariates adjusted for | Male | Female |
|---|---|---|
| Unadjusted | 1.97 (1.46; 2.48) | 1.91 (1.28; 2.55) |
| Vaccination | 1.51 (0.92; 2.10) | 1.19 (0.52; 1.86) |
| Stringency Index | 2.21 (1.70; 2.71) | 2.08 (1.45; 2.71) |
| Trust in government | 1.65 (1.12; 2.18) | 1.68 (1.02; 2.34) |
| Trust in science | 1.75 (1.25; 2.26) | 1.56 (1.05; 2.06) |
| Enforcement of regulations | 1.60 (1.08; 2.12) | 1.44 (0.96; 1.92) |
| Vaccination + Trust in government | 1.08 (0.46; 1.70) | 0.84 (0.09; 1.59) |
| Vaccination + Trust in science | 1.53 (0.98; 2.08) | 1.22 (0.658; 1.780) |
| Vaccination + Enforcement regulations | 1.24 (0.63; 1.85) | 0.84 (0.13; 1.55) |

Note: The Table shows coefficients from nine linear OLS models for the East-West dummy without or with adjustment for covariates. Independent variables: Vaccination – share of fully vaccinated as of the 1st of September 2021; Stringency Index – mean values of the Stringency Index over the weeks of 2021; Trust in government/science – share of people who trust the national government/science reported in 2020; Enforcement regulations – Regulatory Enforcement Index in 2021.

## Discussion

We have found that the differential impacts of the COVID-19 pandemic on excess mortality and life expectancy losses in Eastern compared to Western Europe in 2020−21 have two distinctive features. First is the absence in nearly all Eastern European countries of a pronounced excess mortality in the first few months of the pandemic, unlike in many Western European countries. Second, from October 2020 to the end of 2021, most Eastern European countries tended to experience more persistent high levels of excess mortality compared to Western European countries. This more than offset the approximately 6-month-later onset of substantial excess deaths in Eastern European countries, resulting in a much greater overall toll of excess deaths and life expectancy losses than in Western Europe in the period up to the end of 2021.

The initial low levels of excess mortality in Eastern Europe seen in the first months of the pandemic, when many Western European countries, such as Italy, Spain, and the UK, had massive excess mortality peaks has been noted by others [9–11]. However, little attention has been given to explaining why this might be the case. Countries with low levels of excess mortality in March and April 2020 are likely to be those that had a low prevalence of COVID-19 at the time when they imposed various degrees of restriction and lockdown in March 2020. On the other hand, those with a high prevalence at this point would be more likely to be subject to exponential spread of the virus. Direct estimates of the prevalence of COVID-19 infection have been challenging throughout the pandemic. Several attempts have been made to estimate the prevalence of infection per capita in the early stages of the pandemic. Russell et al. [27] used mortality from COVID-19 to work backwards to estimate the prevalence of infection several weeks earlier, making a series of assumptions about case-fatality and the period from infection to death. In mid-March 2020, when nearly all countries had introduced their first range of public health measures/lockdowns, the estimated prevalence of COVID-19 infection was systematically lower in Eastern compared to Western countries.

As all initial cases in a country of this novel virus will have been the result of importation, the degree of cross-border travel in the immediate pre-pandemic period will be an important driver of early-phase prevalence. One index of connectivity is the number of inbound passenger flights. We have found that there is a positive correlation across countries between the immediate pre-pandemic passenger flight connectivity between European countries and excess mortality in March-April 2020, with the level of flight connectivity and excess mortality being far lower for Eastern compared to Western European countries. The low level of connectivity in the East could have led to a much lower prevalence of COVID-19 cases in March 2020 than in most countries. We interpret this as supporting the hypothesis that in the weeks leading up to the imposition of various degrees of lockdown in March 2020, the scale of the epidemic spread of COVID-19 in each country would be proportional to the number of prevalent cases in the population. If this were low, the imposition of steps to reduce person-to-person transmission would have allowed contact tracing and isolation to flatten the epidemic spread. However, if the number of prevalent cases were high, an epidemic was very likely [28].

In developing a comprehensive explanation of East-West differences in excess deaths and life expectancy losses from COVID-19, it is useful to distinguish between proximal and distal influences (Fig 1). Proximal ones are those that influence the spread of the pandemic in each country, such as promulgation of and population compliance with NPIs and vaccine uptake, including hesitancy and availability, and prevalence of infection at the time of initial lockdown. Distal influences are features that may differ between Eastern and Western European countries that originate in the different post-war histories of the two blocs, which may have influenced how governments and individuals responded to the pandemic. These include differences in public trust in government and science, leading to a varying degree of compliance with government advice or regulation aimed at reducing levels of transmission as well as affecting differences in levels of vaccine hesitancy. We discuss each of these in turn below.

The underlying difference in transport connectivity between East and West can be seen as part of the longer-term legacy of the historical divisions across the European continent in the post-war period. This reflects differences in patterns and intensity of international business and commerce, as well as favoured tourist destinations. The development of major

airline hubs in the UK, France, Germany, Spain, and Italy has occurred over many decades. In Eastern European countries, however, the entry and exit of people was severely constrained up until the fall of the Berlin Wall and the collapse of the Soviet Union.

The summer of 2020 was accompanied by a relaxation of many of the most stringent restrictions on movement and contact between people, both within and between European countries, which drove a resurgence of infections in the early autumn [29]. It would have resulted in the seeding of new infections in Eastern European countries that had been avoided in the first wave of the pandemic, and which eventually drove the substantial peaks of excess mortality in these countries from October 2020 onwards.

In 2021, Eastern European countries lagged those of Western Europe in terms of pace and ultimate level of population coverage of COVID-19 vaccination (S7 Fig). We have found that this is the most influential factor in reducing the size of the East-West difference in life-expectancy losses. Higher levels of vaccination hesitancy in Eastern European countries, particularly in Russia and Bulgaria may be particularly relevant [30–33]. Steinert et al. [34] found no adequate response to messages about the benefits of vaccination in several countries, including Bulgaria and Poland, which was not the case in Germany and the UK.

There has been much scientific and political interest in quantifying the extent to which NPIs played a role in reducing the impact of the pandemic [35–37]. Our analysis found that differences in NPIs as measured by mean levels of the stringency index in each country over 2021 did not explain any of the East-West differences in life expectancy losses in 2021. However, the stringency index we used has many serious shortcomings in this context. Firstly, the relationship between the level of NPIs and SARS-CoV-2 infection was iterative and bidirectional: while the stringency of NPIs would depend on the level of SARS-CoV-2 in the community, the latter would also inform the subsequent level of NPI stringency. Secondly, the stringency index we used summarises the government's intent, rather than actual uptake and compliance. Thirdly, in retrospect, the stringency index was a simple sum of the NPI measures, even though not all the NPIs have the same impact on the transmission of SARS-CoV-2.

What is clear is that the massive fall in cross-border movement that occurred quickly in Europe and elsewhere from March 2020 had a major impact on the spread of the virus [35]. This is the necessary corollary of our conclusion that it was the differences in the prevalence of SARS-CoV-2 infection in European countries that are key to understanding the later excess mortality peaks in Eastern compared to Western countries in 2020.

The extent to which the experience of communism in Eastern Europe in the 20th century has had a persistent negative effect on trust at many different levels of society has been extensively studied [38]. A large multi-regional study conducted in 2020 across 16 countries found that the four countries from Central and Eastern Europe showed the lowest level of trust in government [39]. In our analysis, trust in government, and to a smaller degree, trust in science, appears to explain a part of East-West differences in life expectancy losses in 2021. This could be through a moderating influence on the willingness of individuals to comply with recommended or required behaviour changes aimed at reducing person-to-person transmission. It could also operate through an association of trust with vaccine uptake.

Differences in the degree to which countries in Eastern and Western Europe tend to enforce laws and regulations in general may have an impact on differences in life expectancy losses [17]. This may operate both through the extent of rule-breaking social gatherings as well as business breaches of regulations. The level of trust in institutions as a factor that could influence the impact of the pandemic on populations has been explored in several studies [39–41]. A study looking at factors influencing adherence to NPIs, and its change over time as populations became fatigued, found that reductions in adherence to physical distancing occurred to a smaller degree in countries with high interpersonal trust [42].

Pre-pandemic research on vaccine hesitancy has identified trust as an important factor [43]. This is consistent with the findings of studies of COVID-19 vaccine uptake [44], although another study, specifically of Eastern Europe, using a questionable measure of trust, was inconclusive [45]. However, our results suggest that the combination of vaccine coverage together with trust in government reduces by half the observed East-West differences in life expectancy losses, indicating

that trust in the government may be important above and beyond its relationship to vaccine coverage, possibly through its influence on, among other factors, compliance with NPIs.

Our analysis has weaknesses. Due to the ecological design of the evidence of links between the life expectancy losses and explanatory variables, the corresponding findings are suggestive rather than conclusive. Low statistical power because of the limited number of units of analysis (countries) is also an important concern.

The analysis of flight connectivity was based on the numbers of flights, rather than actual passenger numbers, which would provide the most appropriate metric for assessing the probability of case import. The lack of data on passengers also meant that we could not differentiate between people who landed and entered the destination population versus those who were in transit and traveled onwards to another country. This latter limitation may explain why Germany is such an outlier in S3 Fig. Frankfurt is known for having one of the world's largest transit hubs by passenger volume. Finally, the cross-border spread of infection is a function of land transport and not just flying, although train, bus, and car travel would predominantly be between adjacent countries.

Our connectivity analysis is limited to the intensity of flights considered as a reasonable proxy for the general transport connectivity. However it should be noted that, ground transportation, especially by train, could play an additional role, especially in countries of Central Europe. The variables we used as proxies for the factors that might underlie East-West differences in losses in 2021 may only partially capture the underlying concepts. In addition, we do not have proxy measures for some of the key pathways shown in the conceptual diagram, including the availability and effectiveness of treatments for people with COVID-19. Moreover, we did not have measures of individual and institutional behaviour change that occurred in response to government advice/stipulation or simply individual responses to information available through the news and social media about the pandemic. However, we have used more upstream (pre-pandemic) factors, such as general tendency for enforcement as well as trust in government and science, which we have regarded as potentially important general influences on behaviours of individuals and institutions.

Our conceptual model of life expectancy losses in 2021 does not consider the fact that there are very likely to have been differences between Eastern and Western European countries in the levels of immunity to SARS-CoV-2. Data on population-based seroprevalence of antibodies in Europe at the start of 2021, or at any other point in the pandemic, is very limited [46]. However it is notable that in England, which had one of the earliest and most intense peaks in COVID-19 excess deaths, the prevalence of immunity in the population in mid-July 2020 was surprisingly low at around 6% [47]. Other time series data from the UK show that the proportion of the population with levels of immunity only started to increase above this level once mass vaccination began in early 2021 [48]. However, it is notable that in several Western European countries which had late first peaks in autumn 2020 (e.g., Austria, Switzerland, Germany, Denmark), like all Eastern countries, nevertheless ended up with far lower excess mortality and life expectancy losses in 2021. Overall, we conclude that low rates of naturally acquired immunity at the start of 2021 would be unlikely to explain much of the East-West difference in life expectancy losses in 2021.

## Conclusion

In conclusion, the main early impact of the pandemic in the spring of 2020 was felt in several Western European countries, with none of the former communist countries of Eastern Europe experiencing a major peak in excess mortality until the autumn of 2020. Despite this, the overall level of losses up to the end of 2021 in life expectancy and excess death rates were far higher in the East compared to the West. We argue that both distinctive features of the East-West difference can be ultimately related to the differences in the post-war history of the two blocs. Of particular importance is the lower degree of international transport connectivity of Eastern countries which is likely to explain the lower prevalence of infection in March 2020 when lockdowns were initiated around the world. Lower levels of trust in government and science, and willingness to enforce laws and regulations in the East are likely to be behind the far higher excess mortality rates seen in Eastern than in Western countries in 2021. This appears to have been importantly mediated principally through lower levels of vaccine coverage as well as population adherence to NPIs.

Finally, although not the primary focus of this paper, it is notable that the increase in life expectancy in 2022−23 was much steeper in the East than in the West of Europe. Examination of the mortality changes after 2021 deserves further study. Analysis of the recovery should consider a different set of potential driving forces, such as a stronger "harvesting" effect in the East due to preceding mortality elevations and potentially a higher level of immunity in the East due to higher incidence of COVID-19.

## Supporting information

**S1 Appendix. Mortality data.**
(PDF)

**S2 Appendix. Calculations from weekly mortality data: the weekly death rates, the timing of the earliest major peaks, and the total excess death rate in 2020−21.**
(PDF)

**S3 Appendix. Calculations from weekly mortality data: statistical associations between the excess death rates in March-April 2020 and the flight connectivity across countries in February 2020.**
(PDF)

**S4 Appendix. Calculations from annual mortality data by age and sex: the life expectancy losses in 2020−21 and their split by broad age group.**
(PDF)

**S5 Appendix. Calculations from annual mortality data by age and sex: contributions of relative mortality excess and baseline mortality to the total East-West differences in the life expectancy losses in 2021.**
(PDF)

**S6 Appendix. Statistical associations between the East-West differences in life expectancy losses in 2021 and explanatory variables across countries.**
(PDF)

**S1 Fig. A graphic presentation of different approaches to estimating baseline mortality and corresponding life expectancy losses in 2020–2021: an example of Russia, Poland, and Germany.**
(PDF)

**S2A Fig. Trends in life expectancy at birth in 2000–2023, by sex.**
(PDF)

**S2B Fig. Trends in life expectancy at birth (2000–2023) and the gap in life expectancy between West and East, by sex.**
(PDF)

**S3 Fig. Correlation between the excess death rate in March-April (weeks 10–18), 2020, and the daily number of arriving flights for the period February 10–23, 2020.**
(PDF)

**S4 Fig. Associations between life expectancy losses and predicted life expectancies in 2020 and 2021, by sex.**
(PDF)

**S5 Fig. East-West contrasts in baseline mortality (upper panels) and relative (ratio observed/ baseline) mortality excess, by age in 2021.**
(PDF)

**S6 Fig. Associations between sex-specific life expectancy losses in 2021 with single explanatory factors across countries.**
(PDF)

**S7 Fig. Cumulative prevalence of full vaccination by weeks of 2021 across countries and country groups East and West.**
(PDF)

**S1 Table. Populations and regional groups under study.**
(PDF)

**S2 Table. Mean number of flights per day to each country for the period 10 February to 23 February 2020.**
(PDF)

**S3 Table. Mean, maximal, and minimal life expectancy (LE) losses (95% CI) in groups West and East in 2020 and 2021, in years.**
(PDF)

**S4 Table. Contribution of relative mortality excess and baseline mortality to the East-West difference in life expectancy losses in 2021.**
(PDF)

## Acknowledgments

We would like to thank Professor John Edmunds for his valuable insights and suggestions on an earlier draft of the paper.

## Author contributions

**Conceptualization:** Vladimir M. Shkolnikov, Sergey Timonin, David A. Leon.

**Data curation:** Vladimir M. Shkolnikov, Sergey Timonin, Dmitri Jdanov, Naomi Medina-Jaudes.

**Formal analysis:** Vladimir M. Shkolnikov, Sergey Timonin, Dmitri Jdanov.

**Methodology:** Vladimir M. Shkolnikov, Sergey Timonin, Dmitri Jdanov.

**Visualization:** Vladimir M. Shkolnikov, Sergey Timonin.

**Writing – original draft:** Sergey Timonin, David A. Leon.

**Writing – review & editing:** Vladimir M. Shkolnikov, Sergey Timonin, Dmitri Jdanov, Nazrul Islam, David A. Leon.

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
