## [Decision Letter · Decision Letter 0]

23 Sep 2025

Dear Dr. Timonin,

Thank you for submitting your manuscript to PLOS ONE. After careful consideration, we feel that it has merit but does not fully meet PLOS ONE’s publication criteria as it currently stands. Therefore, we invite you to submit a revised version of the manuscript that addresses the points raised during the review process.

We look forward to receiving your revised manuscript.

Kind regards,

Andrea Nigri

Academic Editor

PLOS ONE

Journal Requirements:

“ST acknowledges support from the Australian Research Council (DP210100401). NI acknowledges support from the UK National Institute for Health and Care Research (HDRUK2022.0313). The funders had no role in study design, data collection and analysis, decision to publish or preparation of the manuscript.”

3. We note that Figure 3 in your submission contain map/satellite images which may be copyrighted. All PLOS content is published under the Creative Commons Attribution License (CC BY 4.0), which means that the manuscript, images, and Supporting Information files will be freely available online, and any third party is permitted to access, download, copy, distribute, and use these materials in any way, even commercially, with proper attribution. For these reasons, we cannot publish previously copyrighted maps or satellite images created using proprietary data, such as Google software (Google Maps, Street View, and Earth). For more information, see our copyright guidelines: http://journals.plos.org/plosone/s/licenses-and-copyright.

a. You may seek permission from the original copyright holder of Figure 3 to publish the content specifically under the CC BY 4.0 license.

Reviewers' comments:

Reviewer's Responses to Questions

**Comments to the Author**

1. Is the manuscript technically sound, and do the data support the conclusions?

Reviewer #1: Yes

2. Has the statistical analysis been performed appropriately and rigorously?

Reviewer #1: Yes

3. Have the authors made all data underlying the findings in their manuscript fully available?

Reviewer #1: Yes

4. Is the manuscript presented in an intelligible fashion and written in standard English?

Reviewer #1: Yes

Reviewer #1: Overall, the manuscript is well written, with a storyline that is well supported by statistical and demographic analysis. I have a few comments that are listed below.

General comments

-I agree with the factor listed, and I found the flight connectivity a very good indicator despite the limitation listed; however, I couldn’t help to think that for the European countries, trains are also an important source of connection, especially within countries. So, I don’t think it is necessary to add this as an additional factor in the analysis, but a couple of sentences in the discussion will be nice.

-In the last paragraph of the conclusions, the author indicated that life expectancy recoveries were larger across Eastern European countries compared to the West after 2021, and that this deserves further investigation. I agree, but, if possible, I would recommend that the authors highlight potential reasons for that.

-Also, given that is a demographic analysis, I missed a discussion about sex differences, or at least some results by sex.

Minor comments

Association between life expectancy losses and factors.

-I agree with the factors and the conceptual framework. But I would ask the authors to explain each indicator, so, for example, what trust enforcement of policies and regulations means, and how it overlaps with policy on non-pharmacological intervention.

-In the supplementary material, they can describe it in more detail and explain what variables are considered for constructing the indices. Especially to know if there is an overlap in the variables used to construct the indices.

-Probably is already in one of those indices, but is there an indicator of the labour market, like the share of individuals working from home, especially for 2021?

Results

-Trends in life expectancy in 2000-2023, I would recommend including an additional figure showing the trends in the gap in life expectancy at birth between East and West to have a clear picture of the trends.

Trends and differences in weekly excess mortality.

-I would avoid the word trends, given that you are selecting a specific month, not changes over time.

Air connectivity and excess mortality

-Can you explain in more detail the rank order correlations?

Figure 4.

-I understand that the colour indicates West or East, and the bars are for 0-64 years old and 65+, but I strongly recommend adding a proper legend.

Discussion

-Very nice discussion.

**Do you want your identity to be public for this peer review?** For information about this choice, including consent withdrawal, please see our Privacy Policy

Reviewer #1: No

---

## [Author Response · Author response to Decision Letter 1]

14 Jan 2026

Please refer to the document “Response to Reviewers” for a detailed table addressing each of the comments.

---

## [Decision Letter · Decision Letter 1]

15 Feb 2026

Widening East-West inequality in life expectancy in Europe during the COVID-19 pandemic: an international comparative study.

PONE-D-25-35060R1

Dear Dr. Timonin,

We’re pleased to inform you that your manuscript has been judged scientifically suitable for publication and will be formally accepted for publication once it meets all outstanding technical requirements.

Kind regards,

Andrea Nigri

Academic Editor

PLOS One

Additional Editor Comments (optional):

Reviewers' comments:

Reviewer's Responses to Questions

**Comments to the Author**

Reviewer #1: All comments have been addressed

2. Is the manuscript technically sound, and do the data support the conclusions?

Reviewer #1: Yes

3. Has the statistical analysis been performed appropriately and rigorously?

Reviewer #1: Yes

4. Have the authors made all data underlying the findings in their manuscript fully available?

Reviewer #1: Yes

5. Is the manuscript presented in an intelligible fashion and written in standard English?

Reviewer #1: Yes

Reviewer #1: The revised version attended all my previous comments, and In general i feel this is a good manuscript that will pick interest among demographer

**Do you want your identity to be public for this peer review?** For information about this choice, including consent withdrawal, please see our Privacy Policy

Reviewer #1: No

---

## [Editor Report · Acceptance letter]

PONE-D-25-35060R1

PLOS One

Dear Dr. Timonin,

I'm pleased to inform you that your manuscript has been deemed suitable for publication in PLOS One. Congratulations! Your manuscript is now being handed over to our production team.

Kind regards,

on behalf of

Dr. Andrea Nigri

Academic Editor

PLOS One